# A CMOS silicon spin qubit

R. Maurand[1,2], X. Jehl[1,2], D. Kotekar-Patil[1,2], A. Corna[1,2], H. Bohuslavskyi[1,2], R. Laviéville[1,3], L. Hutin[1,3], S. Barraud[1,3], M. Vinet[1,3], M. Sanquer[1,2] & S. De Franceschi[1,2]

Silicon, the main constituent of microprocessor chips, is emerging as a promising material for the realization of future quantum processors. Leveraging its well-established complementary metal–oxide–semiconductor (CMOS) technology would be a clear asset to the development of scalable quantum computing architectures and to their co-integration with classical control hardware. Here we report a silicon quantum bit (qubit) device made with an industry-standard fabrication process. The device consists of a two-gate, p-type transistor with an undoped channel. At low temperature, the first gate defines a quantum dot encoding a hole spin qubit, the second one a quantum dot used for the qubit read-out. All electrical, two-axis control of the spin qubit is achieved by applying a phase-tunable microwave modulation to the first gate. The demonstrated qubit functionality in a basic transistor-like device constitutes a promising step towards the elaboration of scalable spin qubit geometries in a readily exploitable CMOS platform.

[1] University Grenoble Alpes, F-38000 Grenoble, France. [2] CEA, INAC-PHELIQS, F-38000 Grenoble, France. [3] CEA, LETI, MINATEC Campus, F-38054 Grenoble, France. Correspondence and requests for materials should be addressed to R.M. (email: romain.maurand@cea.fr) or to S.D.F. (email: silvano.defranceschi@cea.fr).

Localized spins in semiconductors can be used to encode elementary bits of quantum information[1,2]. Spin qubits were demonstrated in a variety of semiconductors, starting from GaAs-based heterostructures[3–5]. In this material, and all III–V compounds in general, electron spins couple to the nuclear spins of the host crystal via the hyperfine interaction resulting in a relatively short inhomogeneous dephasing time, $T_2^*$ (a few tens of nanoseconds in GaAs[6]). This problem can be cured to a large extent by means of echo-type spin manipulation sequences and notch filtering techniques[7–9]. In natural silicon, however, the hyperfine interaction is weaker, being due to the $\approx 4.7\%$ content of $^{29}$Si, the only stable isotope with a non-zero nuclear spin. Measured $T_2^*$ values range between 50 ns and 2 μs (refs 10–14). Experiments carried out on electron spin qubits in isotopically purified silicon (99.99% of spinless $^{28}$Si) have even allowed extending $T_2^*$ to 120 μs (ref. 15). Following these improvements in spin coherence time, silicon-based spin qubits classify among the best solid-state qubits, at the single-qubit level. Recently, the first two-qubit logic gate with control-NOT functionality was also demonstrated[16], marking the next essential milestone towards scalable processors.

Surface-code quantum computing architectures, possibly the only viable option to date, require large numbers (eventually millions) of qubits individually controlled with tunable nearest-neighbour couplings[17,18]. Their implementation is a considerable challenge since it implies dealing with issues such as device-to-device variability, multi-layer electrical wiring and, most likely, on-chip classical electronics (amplifiers, multiplexers and so on) for qubit control and read-out. This is where the well-established complementary metal–oxide–semiconductor (CMOS) technology becomes a compelling tool. A possible strategy is to export qubit device implementations developed within academic-scale laboratories into large-scale CMOS platforms. This approach is likely to require significant process integration development at the CMOS foundry.

Here we present an alternative route, where an existing process flow for the fabrication of CMOS transistors is taken as a starting point and is adapted to obtain devices with qubit functionality. More precisely we define at low temperature, a double quantum dot (QD) inside the channel of a $p$-type silicon transistor with two gates. One QD encodes a hole spin qubit while the other one is used for qubit read-out. We achieve electric field-mediated two-axis coherent control of the hole spin qubit by applying a microwave modulation on one gate of the transistor. Characteristic spin lifetimes ($T_2^*$ and $T_{\mathrm{echo}}$) are revealed by means of Ramsey and spin echo manipulation sequences.

## Results

**Device description.** We use a microelectronics technology based on 300 mm silicon-on-insulator wafers. Our qubit device, schematically shown in Fig. 1a, is derived from silicon nanowire field-effect transistors[19]. It relies on confined hole spins[20–24], and it consists of a 10 nm-thick and 20 nm-wide undoped silicon channel with $p$-doped source and drain contact regions, and two $\approx 30$ nm-wide parallel top gates, side covered by insulating silicon nitride spacers (further details on the spacers are given in Supplementary Note 1). A scanning electron microscopy top view and a transmission electron microscopy cross-sectional view are shown in Fig. 1b,c, respectively. At low temperature, hole QDs are created by charge accumulation below the gates[25]. The double-gate layout enables the formation of two QDs in series, QD1 and QD2, with occupancies controlled by voltages $V_{\mathrm{g1}}$ and $V_{\mathrm{g2}}$ applied to gates 1 and 2, respectively (Supplementary Fig. 2 and Supplementary Note 2). We tune charge accumulation to relatively small numbers, $N$, of confined holes (we estimate $N \approx 10$

and $\approx 30$ for QD1 and QD2, respectively, as discussed in Supplementary Note 2). In this regime, the QDs exhibit a discrete energy spectrum with level spacing $\delta E$ in the 0.1–1 meV range, and Coulomb charging energy $U \approx 10$ meV.

In a simple scenario where spin-degenerate QD levels get progressively filled by pairs of holes, each QD carries a spin $S = 1/2$ for $N =$ odd and a spin $S = 0$ for $N =$ even. By setting $N =$ odd in both dots, two spin-1/2 qubits can be potentially encoded, one for each QD. This is equivalent to the (1, 1) charge configuration, where the first and second digits denote the charge occupancies of QD1 and QD2, respectively. In practice, here we shall demonstrate full two-axis control of the first spin only, and use the second spin for initialization and read-out purposes. Tuning the double QD to a parity-equivalent (1, 1) → (0, 2) charge transition, initialization and read-out of the qubit relies on the so-called Pauli spin blockade mechanism[5,26]. In this particular charge transition, tunnelling between dots can be blocked by spin conservation. Basically, for a fixed, say 'up', spin orientation in QD2, tunnelling will be allowed if the spin in QD1 is 'down' and it will be forbidden by Pauli exclusion principle if the spin in QD1 is 'up', that is, a spin triplet (1, 1) state is not coupled to the singlet (0, 2) state. This charge/spin configuration can be identified through characteristic experimental signatures[27–29] associated with the Pauli blockade effect discussed above (Supplementary Fig. 4 and Supplementary Note 3). (We note that deviations from pairwise filling of the hole QD orbitals can occur, especially beyond the few-hole regime[30], resulting in more complex spin configurations.)

**Electric-dipole spin resonance.** We now turn to the procedure for spin manipulation. In a recent work on similar devices with only one gate, we found that hole $g$-factors are anisotropic and gate-dependent[25], denoting strong spin–orbit coupling[29].

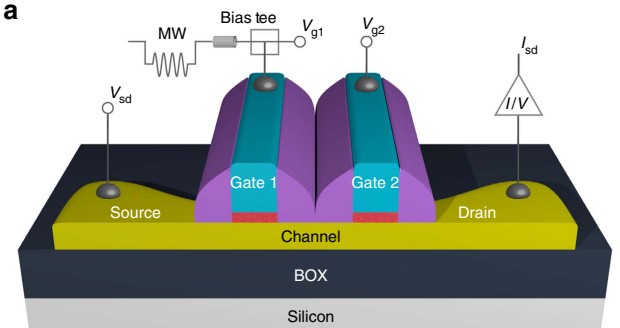

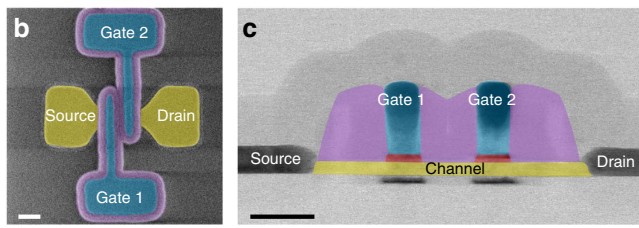

**Figure 1 | CMOS qubit device.** (**a**) Simplified three-dimensional schematic of a silicon-on-insulator nanowire field-effect transistor with two gates, gate 1 and gate 2. Using a bias tee, gate 1 is connected to a low-pass-filtered line, used to apply a static gate voltage $V_{\mathrm{g1}}$, and to a 20 GHz-bandwidth line, used to apply the high-frequency modulation necessary for qubit initialization, manipulation and read-out. (**b**) Colourized device top view obtained by scanning electron microscopy just after the fabrication of gates and spacers. Scale bar, 75 nm. (**c**) Colourized transmission electron microscopy image of the device along a longitudinal cross-sectional plane. Scale bar, 50 nm.

This implies the possibility to perform electric-dipole spin resonance (EDSR), namely to drive coherent hole-spin rotations by means of microwave frequency (MW) modulation of a gate voltage (Supplementary Note 4). Here we apply the MW modulation to gate 1 to rotate the spin in QD1. Spin rotations result in the lifting of spin blockade. In a measurement of source-drain current $I_{sd}$ as a function of magnetic field $B$ (perpendicular to the chip) and MW frequency $f$, EDSR is revealed by narrow ridges of increased current[28]. The data set in Fig. 2a shows two of such current ridges: one clearly visible, most likely associated with QD1 (strongly coupled to the rf-modulated gate); and the other one rather faint, most likely arising from the spin rotation in QD2 (which is only weakly coupled to gate 1). Both ridges follow a linear $f(B)$ dependence consistent with the spin resonance condition $hf = g\mu_B B$, where $h$ is Planck's constant, $\mu_B$ the Bohr magneton and $g$ the hole Landé g-factor (absolute value) along the magnetic field direction. From the slopes of the two ridges we extract two g-factor values $g_1 = 1.63$ and $g_2 = 1.92$ comparable to those reported before[25]. In line with our plausible interpretation of the observed EDSR ridges, we ascribe these g-factor values to QD1 and QD2, respectively. We have observed similar EDSR features at other working points (that is, different parity-equivalent $(1, 1) \rightarrow (0, 2)$ transitions) and in two distinct devices (Supplementary Figs 5 and 6 and Supplementary Note 4).

**Coherent spin control.** To perform controlled spin rotations, and hence demonstrate qubit functionality, we replace continuous-wave gate modulation with MW bursts of tunable duration, $\tau_{burst}$. During spin manipulation, we prevent charge leakage due to tunnelling from QD1 to QD2 by simultaneously detuning the double QD to a Coulomb-blockade regime[4] (Fig. 2b). Following each burst, $V_{g1}$ is abruptly increased to bring the double dot back to the parity-equivalent $(1, 1) \rightarrow (0, 2)$ resonant transition. At this stage, a hole can tunnel from QD1 to QD2 with a probability proportional to the unblocked spin component in QD1 (that is, the probability amplitude for spin-up if QD2 hosts a spin-down state). The resulting $(0, 2)$-like charge state 'decays' by emitting a hole into the drain, and a hole from the source is successively fed back to QD1, thereby restoring the initial $(1, 1)$-like charge configuration. The net effect is the transfer of one hole from source to drain, which will eventually contribute to a measurable average current. (In principle, because not all $(1, 1)$-like states are Pauli blocked, the described charge cycle may occur more than once during the read-out-initialization portion of the same period, until the parity-equivalent $(1, 1) \rightarrow (0, 2)$ becomes spin blocked again and the system is re-initialized for the next manipulation cycle.)

We chose a modulation period of 435 ns, of which 175 ns are devoted to spin manipulation and 260 ns to read-out and initialization. Figure 2c shows an EDSR resonance recorded on

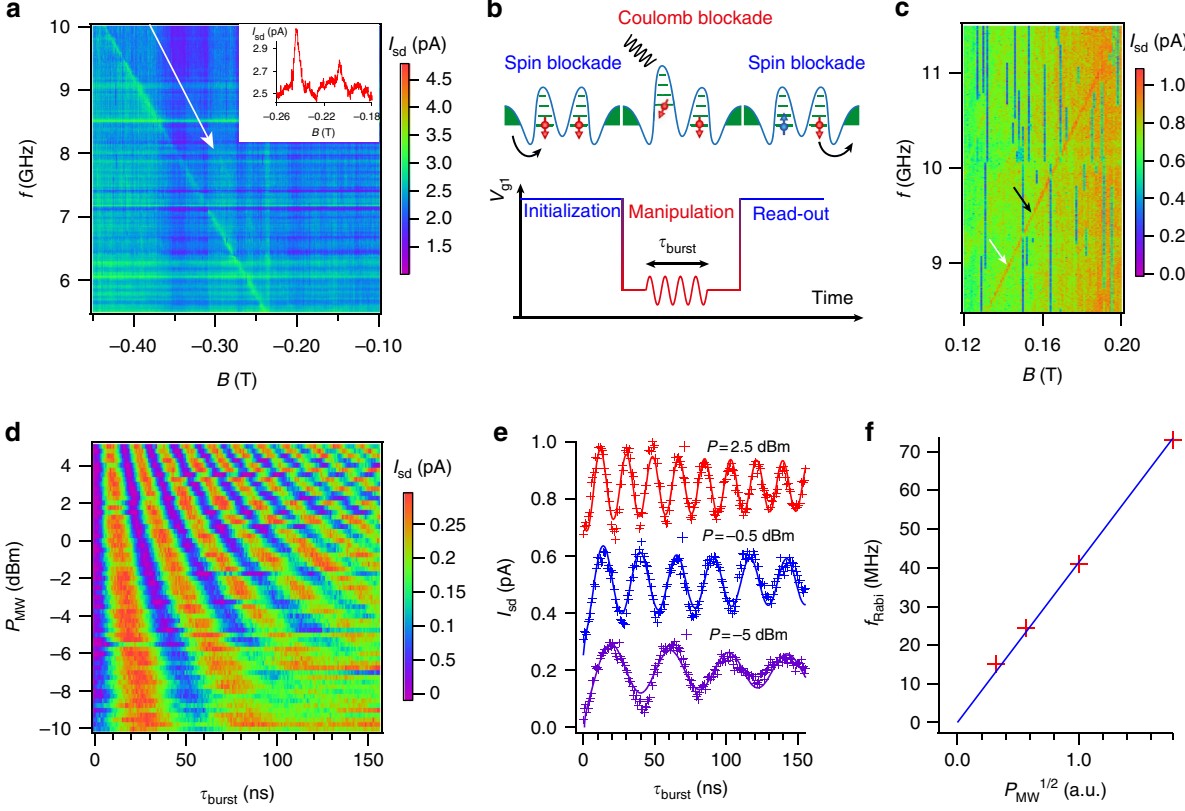

**Figure 2 | Electrically driven coherent spin manipulation.** (**a**) Colour plot of the source-drain current $I_{sd}$ as a function of magnetic field $B$ and MW frequency $f$. Electrically driven hole spin resonance is revealed by two enhanced current ridges. The barely visible upper ridge is indicated by a white arrow. Inset: horizontal cut at $f = 5.4$ GHz. (**b**) Schematic representation of the spin manipulation cycle and corresponding gate-voltage ($V_{g1}$) modulation pattern. (**c**) Same type of measurement as in **a** done on a different device. The cycle presented on **b** is also applied with a MW burst of 20 ns. Coherent manipulations presented in **d**–**f** have been carried at the working point indicated by a white arrow, while the black arrow highlights the working point for Figs 3 and 4. (**d**) Colour plot of Rabi oscillations for a range of microwave powers $P_{MW}$ at $f = 8.938$ GHz and $B = 0.144$ T. (**e**) Rabi oscillations for different powers taken from **c** and fitted (solid lines) to $A \cos(2\pi f_{Rabi}\tau_{burst} + \phi)/\tau_{burst}^{\alpha}$ (ref. 34), current has been averaged for 1 s for each data point. Rabi frequencies are 24, 39 and 55 MHz for $P_{MW} = -5, -0.5$ and 2.5 dBm, respectively. (**f**) Rabi frequency versus microwave amplitude, $P_{MW}^{1/2}$, with a linear fit (solid line).

a second device taken with the previously described gate 1 modulation and a MW burst of 20 ns (a wider $f - B$ range of the EDSR spectrum is shown in Supplementary Fig. 6a). Figure 2d shows $I_{sd}$ as a function of MW power $P_{MW}$, and $\tau_{burst}$ at the resonance frequency for $B = 144$ mT (see white arrow in Fig. 2c). The observed current modulation is a hallmark of coherent Rabi oscillations of the spin in QD1, also explicitly shown by selected cuts at three different MW powers (Fig. 2e). As expected, the Rabi frequency $f_{Rabi}$ increases linearly with the MW voltage amplitude, which is proportional to $P_{MW}^{1/2}$ (Fig. 2f). At the highest power, we reach a remarkably large $f_{Rabi} \approx 85$ MHz, comparable to the highest reported values for electrically controlled semiconductor spin qubits[31]. Figure 3a shows a colour plot of $I_{sd}(f, \tau_{burst})$ revealing the characteristic chevron pattern associated to Rabi oscillations[13]. The fast Fourier transform of $I_{sd}(\tau_{burst})$, calculated for each $f$ value, is shown in the upper panel. It exhibits a peak at the Rabi frequency with the expected hyperbolic dependence on frequency detuning $\Delta f = f - f_0$, where $f_0 = 9.68$ GHz is the resonance frequency at the corresponding $B = 155$ mT (working point indicated by a black arrow in Fig. 2c).

**Dephasing and decoherence times**. To evaluate the inhomogeneous dephasing time $T_2^*$ during free evolution we perform a Ramsey fringes-like experiment, which consists in applying two short, phase coherent, MW pulses separated by a delay time $\tau$. The proportionality between the qubit rotation angle $\theta$ and $\sqrt{P_{MW}}\tau_{burst}$ is used to calibrate both pulses to a $\theta = \frac{\pi}{2}$ rotation (see sketch in Fig. 3c). For each $f$ value, $I_{sd}$ exhibits oscillations at frequency $\Delta f$ decaying on a timescale $T_2^* \approx 60$ ns (Fig. 3b). Extracted current oscillations at fixed frequency are presented in Fig. 3c. At resonance ($\Delta f = 0$), the two pulses induce $\frac{\pi}{2}$ rotations around the same axis (say the $x$ axis of the rotating frame). The effect of a finite $\Delta f$ is to change the rotation axis of the second $\frac{\pi}{2}$ pulse relative to the first one. Alternatively, two-axis control can be achieved also at resonance ($\Delta f = 0$) by varying the relative phase $\Delta\phi$ of the MW modulation between the two pulses. For a Ramsey sequence $\frac{\pi}{2} - \tau - \frac{\pi}{2}_{\Delta\phi}$, the first pulse induces a rotation around $x$ and the second one around $x$, $y$, $-x$ and $-y$ for $\Delta\phi = 0$, $\frac{\pi}{2}$, $\pi$ and $\frac{3\pi}{2}$, respectively. The signal then oscillates with $\Delta\phi$ as shown in the insets to Fig. 4a, and the oscillation amplitude vanishes with $\tau$ on a $T_2^*$ timescale (Fig. 4a).

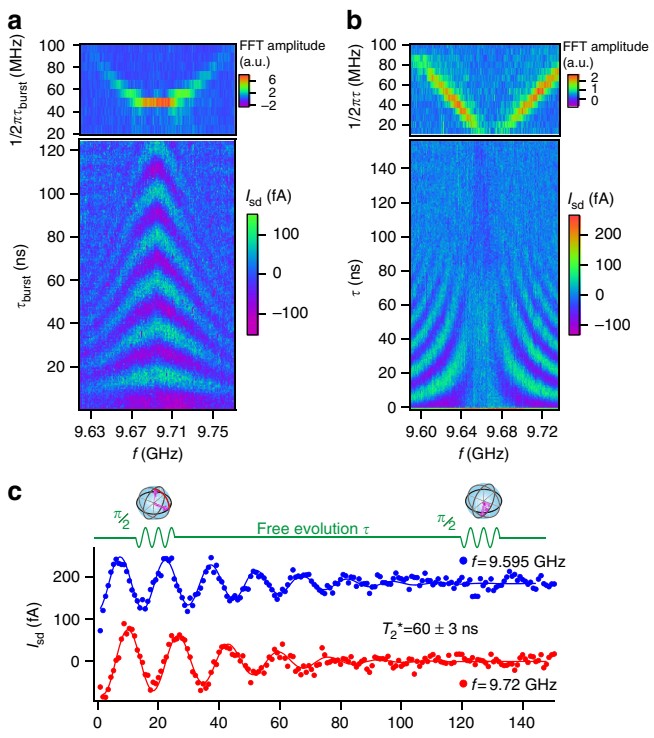

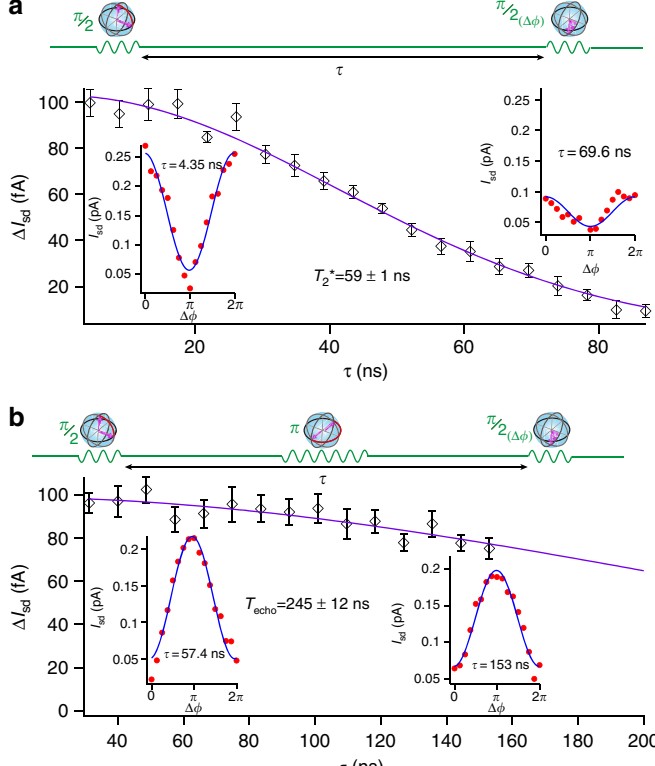

**Figure 3 | Frequency dependence of Rabi oscillations and Ramsey fringes.** (**a**) Bottom panel: $I_{sd}(f, \tau_{burst})$ at $B = 0.155$ T and $P_{MW} = 3$ dBm. Each data point was averaged for 600 ms and, for each $f$, the average current was subtracted. Top panel: Fourier transform of the data in the bottom panel showing the expected hyperbolic dependence of $f_{Rabi}(f)$. (**b**) Bottom panel: $I_{sd}(f, \tau)$, where $\tau$ is the waiting time between two 7 ns-long $\frac{\pi}{2}$ bursts. Each data point was obtained with a 2 s integration time and the average current was subtracted. This data set, taken at $B = 0.155$ T and $P_{MW} = 8$ dBm, shows a characteristic Ramsey-interference pattern. Top panel: Fourier transform of the data in the bottom panel showing the expected linear evolution of the Ramsey fringes frequency. (**c**) Ramsey sequence manipulation scheme (top), and two $I_{sd}(\tau)$ data sets corresponding to vertical cuts in **b** for $f = 9.595$ and $9.720$ GHz. Solid lines are fits to $A\cos(\Delta f\tau + \phi)\exp(-(\tau/T_2^*)^2)$. The data in blue have an upward offset of 200 fA.

**Figure 4 | Two-axis qubit control and spin coherence times.** (**a**) Amplitude $\Delta I_{sd}$ of Ramsey oscillations versus delay time $\tau$. For each $\tau$, the phase of the second $\pi/2$ pulse is shifted by $\Delta\phi$ (see top diagram), which corresponds to a change in the rotation axis. Insets: full $2\pi$ oscillations at short (4.35 ns) and long (69.6 ns) $\tau$ and corresponding sinusoidal fits (solid lines) enabling the extraction of $\Delta I_{sd}$ and associated s.d. error bars. The decay of $\Delta I_{sd}(\tau)$ is fitted to $\exp[-(\tau/T_2^*)^2]$ giving $T_2^* = 59 \pm 1$ ns. (**b**) Results of a Hahn echo experiment, whose manipulation scheme is given in the top diagram. The duration of the refocusing $\pi$ pulse is 14 ns. Insets: full $2\pi$ oscillations at relatively short (57.4 ns) and long (153 ns) $\tau$ and corresponding sinusoidal fits (solid lines). The Hahn echo oscillation amplitude $\Delta I_{sd}$ decays on timescale longer than the largest $\tau$, which was limited to 160 ns to ensure a sufficiently fast repetition cycle, and hence a measurable read-out current. The solid line is a fit to $\exp(-(\tau/T_{echo})^3)$ yielding $T_{echo} = 245 \pm 12$ ns.

Spin echo techniques can extend spin coherence if the source of dephasing fluctuates slowly on the timescale of the hole spin dynamics. We performed a Hahn echo experiment, where a $\pi$ pulse is introduced half way between the two $\frac{\pi}{2}$ pulses, as sketched in Fig. 4b. The amplitude of the oscillations in $\Delta\phi$ (insets to Fig. 4b)) decays on a coherence time $T_{echo} = 245 \pm 12$ ns.

## Discussion

The relatively short $T_2^*$ and $T_{echo}$ (not limited by spin relaxation, see Supplementary Note 5) can hardly be explained by dephasing from $Si^{29}$ nuclear spins. In fact, even if little is known about the hyperfine interaction strength for confined holes in silicon, we would expect it to be even smaller than for electrons[32]. Alternative decoherence mechanisms could dominate, such as paramagnetic impurities, charge noise or the stronger hyperfine interaction with boron dopants diffused from the contact regions. To the best of our knowledge, no relevant magnetic phases should be present in our devices. (Both the nitride-based spacer layers and the silicide contacts are non-magnetic. Magnetic defects may possibly exist in the gate dielectric at the interface between $SiO_2$ and HfSiON but their density should be very low.) Further studies will be necessary to establish statistically relevant values for the coherence timescales and to identify their origin.

In essence, we have shown that a $p$-type silicon field-effect transistor fabricated within an industry-standard CMOS process line can exhibit hole spin qubit functionality with fast, all-electrical, two-axis control. In the prospect of realizing large-scale quantum computing architectures, this result opens a favourable scenario with some clear follow-up milestones. The next step is to advance from the simple, yet limited transistor-like structures studied here to more elaborate qubit designs, incorporating additional important elements such as single-shot qubit read-out (for instance based on rf-gate reflectometry[33]), and enabling scalable qubit-to-qubit coupling schemes. (We refer the reader to Supplementary Note 6 for a more detailed discussion.) In addition, a systematic investigation of qubit performances, including the benchmarking of hole qubits against their electron counterparts has to be performed in the short term. The use of state-of-the-art CMOS technology, with its well-established fabrication processes and integration capabilities, is going to be a clear asset in all these tasks. At a later stage, it should also favour the co-integration of classical cryogenic control hardware.

## Methods

**Device fabrication.** The entire device fabrication process was carried out in a 300 mm CMOS platform. A detailed description is provided in Supplementary Note 1.

**Experimental set-up.** All measurements were performed in a dilution refrigerator with a base temperature of $T = 10$ mK. The direct source-drain current providing qubit read-out was measured by means of a current/voltage amplifier with a gain of $10^9$. All low-frequency lines are low-pass filtered at base temperature with two-stage RC filters. High-frequency signals on gate 1 are applied through a 20 GHz bandwidth coaxial line with 36 dB attenuation distributed along the dilution fridge for thermalization. A home-made bias tee mounted on the sample board enables simultaneous application of microwave and low-frequency signals on gate 1. One channel of an arbitrary wave generator (Tektronix AWG5014C) is used to generate the two-level $V_{g1}$ modulation driving the device between Coulomb blockade (qubit manipulation phase) and Pauli blockade (qubit read-out and initialization). Two other channels of the AWG define square pulses to control the I and Q inputs of the MW source. MW bursts and the two-level gate modulation are combined by means of a diplexer before reaching the dilution fridge.

**Data availability.** The data that support the findings of this study are available from the corresponding authors on reasonable request.

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

## Acknowledgements

We thank D. Estève, M. Hofheinz, F. Kuemmeth, T. Meunier, J. Renard, N. Roch and D. Vion for their help, as well as G. Audoit and C. Guedj for the transmission electron microscopy sample preparation and imaging. The research leading to these

results has been supported by the European Union through the research grants No. 323841, No. 610637 and No. 688539, as well as through the ERC grant No. 280043.

## Author contributions

X.J., M.S., S.D.F., R.L., S.B. and M.V. designed the devices; R.L., L.H., S.B. and M.V. followed their fabrication process; R.L., D.K.-P., A.C. and H.B. characterized the basic electronic properties of the devices; R.M. performed the experiments with input from S.D.F., R.M., X.J., M.S. and S.D.F. analysed the results; R.M. and S.D.F. wrote the manuscript, with input from all authors; S.D.F., X.J. and M.S. initiated the project.

## Additional information

**Competing financial interests:** The authors declare no competing financial interests.

**Publisher's note**: 

