## [Peer Review File · Nature Communications]

Reviewers' comments:

Reviewer #1 (Remarks to the Author)

A. Summary of the key results

The manuscript reports on the spin manipulation experiments of a system of holes in a silicon FinFET with two gates in series. The Authors show Pauli spin blockade, Rabi oscillations and Ramsey fringes of a system of holes of unknown number by averaging current transport measurements.

B. Originality and interest: if not novel, please give references

The manuscript reports experiments similar to those performed in other semiconductor quantum dots including silicon, cited by the Authors, in a novel combination consisting of holes in silicon which is currently not reported yet. The experimental observations carried on the qubits are interesting. The fact that the number of holes is high and unknown strongly affects the impact and the generality of the conclusions.

The Authors claim the use of CMOS fabrication technique as a major breakthrough of their research in the prospect of qubit scalability, but first the system does not make use of complementarity of CMOS, and more importantly the method shows significant limitations in the control of holes in the quantum dots and therefore introduces a degree of strong variability, therefore affecting the claim of potential scalability.

C. Data & methodology: validity of approach, quality of data, quality of presentation

The quality of the Rabi oscillations and Ramsey fringes data is good. The issues arise for what concerns the data to clarify what kind of spin resonance the Authors are observing.

The Authors claim the regime is of about 10 holes per dot, but the data are not shown. First, a Reader should have the opportunity to check the data and make his own idea about. Second, the data are required to support the claim. Such lack of data was already mentioned in the previous report to the Editor with the request of providing the data, but it has been ignored by the Authors in the new submission.

Next, the Authors introduce an oversimplified notation (1,1)-like and (2,0)-like states and they miss to define with what they call "parity equivalent states", they miss to demonstrate how and when this is rigorously applicable in their experiments in place of the more complex condition $(N+1, N+1) \rightarrow (N+2, N)$ where both light and heavy holes may be involved at the same time, making the qubit potentially not workable. Their reply to previous Referee Report that such explanation is too obvious for experts, and too difficult to non-expert is an odd reason to escape the answer, while leaving a gap in their arguments.

The lack of knowledge about the system under investigation, about the involvement of heavy/light hole, about the conditions under which the naive assumptions can be taken make the ground of the experiment very shaky. Such nice experiment, performed under unclarified conditions, provides relatively limited understanding of what is really happening and to enable someone else to reproduce similar conditions to get the same results.

D. Appropriate use of statistics and treatment of uncertainties

The Authors claim they are in the regime of about 10 holes, but they do not provide the uncertainties and how they obtain it.

In their reply to a previous Referee Report, they declare that

"Our estimate of about 10 holes is a rather rough one."

"Therefore our estimate of the number of holes is accurate to a 10 holes uncertainty."

which, if I understand correctly, should be " 10 ± 10 " but neither the data with the method nor the uncertainty evaluation are reported in the manuscript.

E.Conclusions: robustness, validity, reliability

The Authors do not include in the Conclusion section the summary of their findings. The Conclusions section is built as a list of future experiment planned by the Authors.

F.Suggested improvements: experiments, data for possible revision

The improvements have been listed above. Please notice that MW is megawatt, not microwave.

G.References: appropriate credit to previous work?

After adding the references mentioned by the Reviewers in the previous Referee Report to Q 9857H98Q, the previous work is appropriately credited.

H.Clarity and context: lucidity of abstract/summary, appropriateness of abstract, introduction and conclusions

The Authors oversell the employment of a CMOS compatible process. As already explained in a previous Referee Report, the device is not CMOS. As declared by the Authors,

"The device consists of a two-gate, p-type transistor with an undoped channel."

and they consider this a **step** towards CMOS:

"The demonstrated qubit functionality in a basic transistor-like device constitutes a promising step towards the elaboration of scalable spin qubit geometries in a readily exploitable CMOS platform."

Indeed, they conclude:

"we have shown that a p-type silicon field-effect transistor fabricated within an industry-standard CMOS process line can exhibit hole spin qubit functionality"

The title is misleading as it oversells the results by creating the impression that a CMOS qubit is demonstrated, while the experiments are carried on a p-type FinFET where complementarity plays no role. The title should clearly reflect that the device is not CMOS but a p-type FinFET. If, on the contrary the focus is on the industry-standard CMOS process, the number of issues raised by the variability and lack of control on the number of holes associated to the process weakens the claim as the scalability is not proven.

As the main point of the paper is the demonstration of hole qubits with a potentially scalable approach, I observe that the main criticisms contained in the Remark 1 of both Reviewer 2, Reviewer 3 and by myself in the previous Referee Report to Q 9857H98Q is not addressed by adding the Supplementary S6, which sounds too superficial to address the issue of scaling CMOS quantum dots of the manuscript in a real architecture including for instance virtual qubit control necessary to exploit 2-spin states, dynamical decoupling, etc.

To conclude, even if the qubit experiments are a very good piece of work on a complicated physical system, the new version of the manuscript does not fix the major issues raised in the previous Referee Reports. The lack of an adequate theoretical and experimental framework to assess the conditions under which the qubit experiments are carried, the consequent lack of

generality and the interest for a more specialized Readership make the manuscript unsuitable to be published in Nature Communication journal.

Reviewer #3 (Remarks to the Author):

In the revised version of the manuscript by Maurand et al., A CMOS spin qubit in silicon; the authors have satisfactorily addressed my criticisms of the previous version of the document. As stated in my previous review this result is a major milestone in the integration of quantum information technologies with conventional CMOS. The experiment is appealing because using transistors, commonly devoted to processing large amounts of classical information, to perform quantum operations could facilitate the task of scaling silicon-based quantum computation to a large number of qubits and control electronics.

Hence, I recommend the article for publication in Nature Communications.

As I minor recommendation, a mention to multi qubit architectures, already demonstrated in App. Phys Lett. 108 203108, could facilitate the reader the understanding of how CMOS qubits could be scaled up.

Response to Referees:

Comment from Reviewer #1 (point C) and related request from Reviewer #3

We have expanded supplementary section S2 by including a new figure (Fig. S2-2) and related text where we explain how we give an estimate of the number of holes in each quantum dot as requested by reviewer #1. Moreover, following the suggestion from reviewer #3, we have added a stability map around the working point of Fig. 2a).

Comment from Reviewer #3: "This distribution of the figures seems confusing. I would suggest swapping Fig 2.a with FigS4-2 to maintain the coherence of the story so the reader can see the frequency spectrum of the device that was actually used to perform the Rabi oscillations."

We have followed the advice of reviewer #3 and modified Fig. 2 of the main text by adding a panel (panel c)) showing a close-up of Fig. S4-2. The reader can now see the frequency spectrum used to perform the Rabi oscillations directly in the main text. The caption of Fig. 2 has been modified accordingly. References to this new panel appear in the main text lines 121-123, 125 and 135-136.

Comment from Reviewer #3: As a minor recommendation, a mention to multi qubit architectures, already demonstrated in App. Phys Lett. 108 203108, could facilitate the reader the understanding of how CMOS qubits could be scaled up.

We have added this reference to supplementary material S6 where we discuss the scalability prospects.

Comment from Reviewer #3: The authors should note that Pauli-spin blockade has been observed in another report on p-type silicon devices: Li et al. Nano Letters 2015, 15 (11), pp 7314–7318

This reference is already present in the main manuscript (reference 28).